# Intra-Individual Variability of Human Dental Pulp Stem Cell Features Isolated from the Same Donor

**DOI:** 10.3390/ijms222413515

**Published:** 2021-12-16

**Authors:** Nela Pilbauerova, Jan Schmidt, Tomas Soukup, Jan Duska, Jakub Suchanek

**Affiliations:** 1Department of Dentistry, Charles University, Faculty of Medicine in Hradec Kralove and University Hospital Hradec Kralove, Sokolska 581, 500 05 Hradec Kralove, Czech Republic; nela.pilbauerova@lfhk.cuni.cz (N.P.); duskaj@lfhk.cuni.cz (J.D.); suchanekj@lfhk.cuni.cz (J.S.); 2Department of Histology and Embryology, Charles University, Faculty of Medicine in Hradec Kralove, Simkova 870, 500 03 Hradec Kralove, Czech Republic; soukupto@lfhk.cuni.cz

**Keywords:** dental stem cells, mesenchymal stem cells, regenerative medicine, intra-individual variability, same donor isolation, stem cell characterization

## Abstract

It is primarily important to define the standard features and factors that affect dental pulp stem cells (DPSCs) for their broader use in tissue engineering. This study aimed to verify whether DPSCs isolated from various teeth extracted from the same donor exhibit intra-individual variability and what the consequences are for their differentiation potential. The heterogeneity determination was based on studying the proliferative capacity, viability, expression of phenotypic markers, and relative length of telomere chromosomes. The study included 14 teeth (6 molars and 8 premolars) from six different individuals ages 12 to 16. We did not observe any significant intra-individual variability in DPSC size, proliferation rate, viability, or relative telomere length change within lineages isolated from different teeth but the same donor. The minor non-significant variances in phenotype were probably mainly because DPSC cell lines comprised heterogeneous groups of undifferentiated cells independent of the donor. The other variances were seen in DPSC lineages isolated from the same donor, but the teeth were in different stages of root development. We also did not observe any changes in the ability of cells to differentiate into mature cell lines—chondrocytes, osteocytes, and adipocytes. This study is the first to analyze the heterogeneity of DPSC dependent on a donor.

## 1. Introduction

Dental pulp stem cells (DPSCs) are adult multipotent mesenchymal stem cells (MSCs) [1]. As such, they are capable of differentiation in mesodermal or neuroectodermal cells, including odontoblasts, osteoblasts, chondrocytes, myocytes, neurocytes, hepatocytes, melanocytes, or cells producing insulin [2,3,4,5]. DPSCs have advantageous features in comparison with stem cells isolated, for example, from bone marrow [6], skin [7], or peripheral blood [8]. The uniqueness of DPSCs mainly lies in the ease of accessibility, high proliferative potential, and broad cell spectrum multipotency [9]. Their high plasticity and multipotential capacity to differentiate into a large array of tissues can be explained by their neural crest origin [10]. The potential usage of DPSCs in regenerative or reparative medicine is not limited to the field of dentistry [11]. Beneficial features of DPSCs have been demonstrated, for example, in central and peripheral nerve cell regeneration and repair after mechanical or ischemic damage [12,13,14,15,16], in accelerated repair of neuro-ophthalmic nerves and the eye [17,18,19,20], repair of bone or cartilage [21,22,23], or regeneration of liver fibrosis [24].

One DPSC source is the dental pulp of permanent teeth. The dental pulp is the soft connective tissue of the tooth containing four layers. The external layer (odontoblast layer) is made up of odontoblasts producing dentin; the second layer (cell-free zone) is poor in cells and rich in collagen fibers; the third layer (cell-rich zone) contains fibroblasts and dental pulp stem cells. From this layer, undifferentiated stem cells migrate to various districts where they can differentiate under different stimuli and make new differentiated cells and tissues, such as a “dentin-like tissue” (a reparative or tertiary dentin). The innermost layer (core of the pulp) comprises blood vessels and nerves that enter the tooth mostly through the apical foramen. Other cells in the pulp include fibrocytes, macrophages, and lymphocytes [25]. The dentin and enamel act as barriers, separating the pulp tissues from external differential stimuli; pulp environment structures called niches help keep the DPSCs in their primitive nature. Another advantage that DPSCs have over other adult mesenchymal stem cells is the ease of harvesting. The most frequently used source of DPSCs is the dental pulp from the extracted third molars [26]. Therefore, the isolation of stem cells for future treatment can be viewed as a part of a planned extraction procedure performed under local anesthesia rather than an unnecessary intervention performed only for stem cell isolating. DPSC isolation is easy, fast, not financially demanding, and safe [10].

However, many questions must be addressed concerning the possible broader use of DPSCs in tissue engineering and regenerative medicine. It is primarily important to define the standard DPSC characterizations and the factors that affect them. The Mesenchymal and Tissue Stem Cell Committee of the International Society for Cellular Therapy proposed minimal criteria to define human MSCs [27]. First, such cells must be plastic-adherent when undergoing culturing in standard cultivation conditions. Second, they must express several clusters of differentiation markers, and third, MSCs must differentiate into osteoblast, adipocytes, and chondroblasts in vitro [27]. Previous research has described various properties of DPSCs in relation to the method of isolation, cultivation [28], and the inter-individual, age-dependent variances among donors [29,30]. Some research studies have focused on comparing features of stem cells isolated from various tissue sources, including dental pulp [31,32,33].

None of the published studies have investigated the diversity of DPSCs isolated from different teeth extracted from the same donor. The main question of this study is whether DPSC phenotypes exhibit intra-individual variability (and if so, to what extent) and what role this might play in the differentiation potential. With this in mind, our study seeks to compare the basic phenotypic characteristics of DPSCs (isolated from the same donor and cultured under the same conditions) and to define potential differences concerning their ability to differentiate into osteoblasts, chondroblasts, and adipocytes.

## 2. Results

### 2.1. Donor and Tooth Overview

The study included 14 teeth from six different individuals ages 12 to 16 (Table 1).

Teeth isolated from donors 1, 2, 3, 5, and 6 were extracted during one appointment on the same day. Extracted premolars obtained from donor 4 were extracted in two sessions. Teeth from donors 1–5 were extracted under local anesthesia, but molars from donor 6 were extracted under general anesthesia.

### 2.2. Laboratory Procedures

After approximately seven days of initial seeding, adherent fibroblast-like spindle colonies were observed. Once they reached confluence, passaging of the adherent cells resulted in rapid multiplication. All DPSCs were cultivated up to the 8th passage. Paired lineages of DPSCs isolated from the same donor showed a similar proliferation rate (Figure 1 and Figure 2). A slight non-significant variance was observed in donors 1, 2, and 6. All these extracted teeth were molars. However, lineage 1-A was isolated from tooth 27 and 1-B from tooth 28. The same example was seen in donor 6, where lineage 6-A was isolated from tooth 37 and 6-B from tooth 38. In both cases, the wisdom teeth were in more immature root development than the second molars.

During each passaging, we also measured cell diameter. Figure 3 illustrates the average cell size during cell cultivation.

The cell viability was carried out using a Vi-Cell analyzer (Beckman Coulter, Miami, FL, USA) in the 2nd and 8th passages. All paired DPSC lineages contained more than 89% viable cells. Lineage 2-B (isolated from tooth 28) had the highest percentage of viable cells in both measurements. However, the cell viability of all paired lineages did not differ statistically (Figure 4).

The DPSCs are identified by specific marker gene expressions. The minimal criteria set by the International Society for Cellular Therapy assure MSC identity by using CD105, CD73, and CD90, and lack expression of CD45, CD34, CD14, or CD11b, CD79alpha or CD19 and HLA-DR surface molecules [34]. To verify whether there is significant intra-individual variability in phenotype patterns, we performed the flow cytometry analysis in the 3rd and 7th passages of all paired lineages of DPSCs. We did not observe a statistical difference in expressions of any of the analyzed CD markers (Figure 5). All paired lineages highly (>70%) expressed CD markers for mesenchymal stem cells: CD29, CD44, CD73, CD90, or stromal-associated stem cell markers CD13 and CD166. They showed no (<10%) or low marker gene expressions (<40%) of CD31 (platelet endothelial cell adhesion molecule (PECAM-1), CD34 (a transmembrane phosphoglycoprotein for hematopoietic stem cells), and CD45 (protein tyrosine phosphatase for hematopoietic stem cells).

To determine whether there is a difference in the relative telomere length of DPSCs isolated from different teeth but the same donor, we performed a quantitative PCR assay in different passages (2nd passage and 7th passage). The analysis results are depicted in Figure 6. Although we observed the variance between lineages of donor 2, there was no statistically significant difference among paired lineages. Lineage 2-B prolonged its relative telomere length in the 7th passage in comparison with lineage 2-A. However, the overall trend among all lineages was to shorten the relative telomere length with increasing passage number. This was statistically significant (*p* = 0.002).

### 2.3. Differentiation Assay

We did not prove the hypothesis that there are consequences of intra-individual variability in DPSC phenotypes in terms of their differentiation potential. We were successfully able to trigger osteogenesis, chondrogenesis, and adipogenesis in all paired lineages of DPSCs according to external cultivation conditions. To verify the success of differentiation we performed immunocytochemical detection (type II collagen and osteocalcin) and histological staining (blue Masson trichrome, von Kossa staining, and Oil Red-O staining). Results are illustrated in the following figures (Figure 7, Figure 8, Figure 9, Figure 10 and Figure 11). We chose representative examples of paired lineages (5-A and 5-B).

## 3. Discussion

Previously published studies have described the basic properties of DPSCs and how they vary depending on the different methods of isolation or cultivation [28,35,36] or the donor age [29,30]. There are also studies describing various characterizations of dental stem cells isolated from different tissues, including DPSCs [31,32,37]. This study aimed to verify whether DPSCs isolated from various teeth extracted from the same donor exhibit intra-individual variability and what the consequences are for their differentiation potential. Significant intra-individual heterogeneity among DPSCs isolated from various teeth of the same donor would reveal the potential issues that should be taken into consideration during planning the research methodology. Conversely, the absence of intra-individual heterogeneity would simplify laboratory practices. In both cases, the conclusion of our study will have an impact on the standardization of laboratory protocols. Mehrabani et al. published a study showing that DPSCs collected from different teeth showed different properties, especially different proliferation rates [38]. However, it was based on a comparison of dental pulp stem cells from different donors. Although the results of that study are valuable, they are nonetheless limited in relation to the different properties of DPSCs because they were dependent on the type of tooth from which the cells were isolated.

Therefore, our study aimed to compare the basic phenotype characteristics of DPSCs isolated from the same donor and under the same cultivation conditions and to define the potential differences in multipotency. First, we wanted to verify whether there is a significant difference in the DPSC phenotypes within paired lineages isolated from the same donor. To answer this question, we compared paired DPSC lineages in proliferation capacity, viability, and phenotype profile typical for mesenchymal stem cells using flow cytometry and determined the relative telomere length change in different passages during cultivation. Second, we studied whether the intra-individual variability in basic DPSC characteristics affected the ability to differentiate into osteocytes, chondrocytes, and adipocytes.

The study included 14 teeth from six different individuals ages 12 to 16. The young age of patients resulted from the fact that the most common reason for extraction was an initiation of orthodontic treatment during which more than one tooth was extracted. Regarding the characterization of DPSCs, we did not observe any significant changes in the proliferation capacity as determined by population doubling time (PDT) or cumulative population doublings (PD). The biggest differences were seen within lineages isolated from donors 1 and 6. Lineage 1-A was isolated from tooth 27 and 1-B from tooth 28. The same example was seen in donor 6, where lineage 6-A was isolated from tooth 37 and 6-B from tooth 38. In both cases, the wisdom teeth were in more immature root development than the second molars. We also observed a difference in donor 2, but in this case, both extracted teeth were wisdom teeth. However, lineage 2-B was the only one that did not exhibit the relative telomere length shortening with increasing passaging. Other lineages displayed shortened telomere chromosomes with increasing passaging. Extensive in vitro proliferation of human DPSCs is associated with telomere attrition [39,40]. Regarding the phenotype profile, all paired lineages highly proliferated the mesenchymal stem cells markers and showed no or low positivity for hematopoietic or endothelial markers. However, we found particular differences in expressions of CD markers between cell lineages isolated from the same donor. None of them were statistically significant. We suppose that DPSC cell cultures comprise different types of undifferentiated cells and that this heterogenicity is probably donor-independent. Furthermore, we investigated whether the intra-individual heterogeneity affected the multipotency of isolated lineages. We successfully triggered osteogenic, chondrogenic, and adipogenic differentiation in all paired lineages. We verified our findings using immunocytochemistry to reveal osteocalcin and type II collagen. We also stained samples using histological staining (von Kossa staining, blue Masson trichrome, and Oil Red-O) to detect calcium phosphate deposits, collagen and procollagen, adipose vacuoles, and droplets.

We are fully aware of the study limitations. First, we had only two paired lineages isolated from different types of teeth (donor 1 and 6). For a wider range of results, it will be necessary to study DPSCs isolated from different types of teeth extracted from the same donor. However, it seems thus far that the intra-individual variety between DPSCs isolated from the same donor is dependant on the tooth stage development. Fewer variations meant that DPSC cell cultures comprised different types of undifferentiated cells; such heterogenicity was probably independent of the donor. Second, it would be better to quantify the differentiation ability within the lineages isolated from the same donor using the PCR method to evaluate particular differentiation markers. In our study, we only verified their ability to differentiate into three different mature cell lines.

In our future studies we would like to analyse paired lineages isolated from the same donor, but different tooth types (premolars vs. molars or other combinations). We also would like to quantify the differentiation ability within the lineages isolated from the same donor.

## 4. Materials and Methods

### 4.1. Donors

All donors and/or their legal representatives were informed of the purpose of our study and gave informed consent before being included in the study. The Ethical Committee of University Hospital Hradec Kralove approved the study guidelines and informed consent (ref. no. 352 201812 SO7P). Inclusion criteria were the extraction of at least two teeth. Exclusion criteria were carious or periodontally compromised teeth. Common reasons for tooth extraction were recurrent inflammatory complications of soft tissues around semi-impacted third molars, ectopic localization of impacted third molars, or premolar extraction as a part of ongoing orthodontic treatment. All patients were healthy individuals with no history of smoking.

### 4.2. DPSCs Isolation

Immediately after the extraction, each tooth was cleaned using a sterile gaze to remove the microbial plaque. Afterward, the teeth were decontaminated using 0.2% solution of chlorhexidine gluconate for 30 s. After this step, extracted teeth were placed in tubes with a chilled transportation medium containing 1 mL of Hank’s balanced salt solution (HBSS, Invitrogen, Carlsbad, CA, USA), 9 mL water for injection (Bieffe Medital, Grosotto, Italy), and antibiotics and an antifungal agent against any potential contamination, including 200 μL/10 mL streptomycin (Invitrogen), 200 μL/10 mL gentamicin (Invitrogen), 200 μL/10 mL penicillin (Invitrogen), and 50 μL/10 mL amphotericin (Sigma-Aldrich Co., St. Louis, MO, USA). The temperature was kept at 4 °C during transportation to the tissue laboratory at the Department of Histology and Embryology at the College of Medicine in Hradec Kralove.

In the laboratory, samples were proceeded inside the culture room in a laminar flow chamber on the same day as the tooth extractions were done. In the beginning, a pulp chamber of teeth was revealed by splitting the tooth in a cement enamel junction using Luer forceps. After opening the pulp chamber, we removed the pulp tissues using a sharp probe and tweezers. After the pulp tissues were retrieved, we minced them using sterile scissors, ground them using a mini-tissue grinder with an isotonic solution (phosphate-buffered saline, PBS, (Sigma-Aldrich), and finally dissociated them enzymatically using 0.05% Trypsin-EDTA (Gibco, London, UK) for 10 min at 37 °C. After this period, the enzymatic digestion was neutralized using a neutralization medium (20% of Alpha-MEM cultivation medium (Gibco) and 80% fetal bovine serum (FBS, PAA Laboratories, Inc., Dartmouth, MA, USA).

After centrifugation (600× *g*, 5 min), the cell pellet was resuspended in a modified cultivation medium Minimum Essential Medium Eagle—Alpha modification: (Alpha-MEM, Gibco) for mesenchymal adult progenitor cells containing 2% fetal bovine serum (FBS, PAA Laboratories) and supplemented with 10 ng/mL epidermal growth factor (PeproTech, London, UK), 10 ng/mL platelet-derived growth factor (PeproTech), 50 nM dexamethasone (Sigma-Aldrich), 0.2 mM L-ascorbic acid (Bieffe Medital) for protection against oxygen radicals, essential amino acid glutamine (Invitrogen) at a final concentration of 2%, and antibiotics—100 U/mL penicillin, 100 µg/mL streptomycin (Invitrogen), 20 μg/mL gentamicin (Invitrogen), and 0.4 µL/mL amphotericin (Sigma-Aldrich). The culture medium was changed every 3 days; prior to this, the tissue culture dish was washed with PBS to remove non-adherent elements and detritus. We kept the cultivation dishes at a temperature of 37 °C and at 5% CO_2_. We reviewed the culture dishes regularly and, after the cells reached 70% confluence, we passaged them using 0.05% Trypsin-EDTA and resuspended them in a final concentration of 5000 cells/cm^2^. All lineages were passaged up to the 8th passage (8p).

### 4.3. Laboratory Procedures

To find out whether there is significant intra-individual variability, we compared DPSC characteristics based on proliferative capacity, viability, expression of phenotypic markers typical for mesenchymal stem cells, and the relative length of telomere chromosomes in different passages.

#### 4.3.1. DPSC Size, Proliferation, and Viability

During each passaging, we analyzed DPSC diameters and the total DPSC count using Z2-Counter (Beckman Coulter). The protocol of measurement was according to manufacturer’s instructions. After each cell passaging, centrifugation, the cell pellet was resuspended in the 1 mL of the culture medium and 100 µL of cell suspension mixed with 9.9 mL of diluent was used for analysis. The Z2-Counter analyzer is based on the detection and measurement of changes in electrical resistance produced by cells suspended in a conductive liquid (diluent) traversing through a small aperture. Proliferation activity was determined in each passage as cumulative population doublings (PDs) and population doubling time (PDT). The formula for these measurements is described in our previous study [40]. Viability assay was performed using the Trypan Blue Dye Exclusion method in the 2nd and 8th passages. Equal volumes of cell suspension and trypan blue were mixed and automatically analyzed using a Vi-Cell analyzer (Beckman Coulter).

#### 4.3.2. Flow Cytometry

In order to verify intra-individual variety in the pattern of clusters for differentiation markers, we used immunophenotyping against the following markers: CD10 (CB-CALLA, eBioscience, San Diego, CA, USA), CD13 (WM-15, eBioscience), CD18 (7E4, Beckman Coulter), CD29 (TS2/16, BioLegend, San Diego, CA, USA), CD31 (MBC 78.2, Invitrogen), CD34 (581 (Class 287 III), Invitrogen), CD44 (MEM 85, Invitrogen), CD45 (HI30, Invitrogen), CD49f (GoH3, Invitrogen), CD63 (CLBGran/12, Beckman Coulter), CD73 (AD2, BD Biosciences Pharmingen, Erembodegen, Belgium), CD90 (F15-42-1-5, Beckman Coulter), CD105 (SN6, 289, Invitrogen), CD106 (STA, BioLegend), CD117 (2B8, Chemi-Con, Nuremberg, Germany), CD146 (TEA1/34, Beckman Coulter), CD166 (3A6, Beckman Coulter), CD271 (ME20.4, BioLegend), MHC class I (Tu149, Invitrogen), MHC class II (Tü36, Invitrogen). For flow cytometric analysis, cells were detached and stained sequentially with primary immunofluorescence antibodies conjugated with phycoerythrin (PE) or fluorescein (FITC) against the above-mentioned CD markers before a Cell Lab Quanta analysis (Beckman Coulter). The percentage of positive cells was determined as a percentage of cells with higher fluorescence intensity than the upper 0.5% isotype immunoglobulin control. Classification criteria were as follows: <10% no expression, 10–40% low expression, 40–70% moderate expression, and >70% high expression [41].

#### 4.3.3. Quantitative PCR

To determine the relative telomere length, we used a previously described method [39,40]. Briefly, telomere length measurement was performed by qPCR assay. We extracted the DNA of isolated stem cells using a DNeasy Tissue Kit (Hilden, Germany). After the DNA isolation, we calculated its concentration in each sample using a spectrophotometer Nanodrop 1000 (Thermo Fisher Scientific, Waltham, MA, USA). The relative telomere length was calculated using the formula T/S = 2^−ΔCt^, where ΔCt = Ct_telomere_ − Ct“_single copy_” gene. The single gene (housekeeping gene) was a coding acidic ribosomal phosphoprotein 36B4. We performed the qPCR in 96-well plates, and we analyzed each sample in triplicates at the same well position on an ABI 7500 HT detection system (Applied Biosystems, Foster City, CA, USA). Each 20 μL reaction consisted of 20 ng DNA, 1 × SYBR Green master mix (Applied Biosystems), 200 nM forward telomere primer (CGG TTT GTT TGG GTT TGG GTT TGG GTT TGG GTT), and 200 nM reverse telomere primer (GGC TG TCT CCT TCT CCT TCT CCT TCT CCT TCT CCT). We used the following primer pairs for the housekeeping gene analysis: 36B4u, CAG CAA GTG GGA AGG TGT AAT CC; 36B4d, CCC 135 ATT CTA TCA TCA ACG GGT ACA A. The standard of DNA quantum was verified using one reference sample diluted to these final concentrations: 0.02, 0.20, and 2.00 ng/μL. The cycling of each qPCR analysis (for both telomere and housekeeping gene) started with a ten-minute cycle at 95 °C, followed by 15-s cycles at 95 °C, ending with a one-minute cycle at 60 °C. We analyzed the difference in the relative telomere length between the 2nd and 7th passages.

### 4.4. Differentiation Assay

To verify the hypothesis that the intra-individual variability of DPSC phenotypes affects the ability to differentiate into osteocytes, chondrocytes, and adipocytes, we triggered osteogenesis, chondrogenesis, and adipogenesis in isolated stem cells.

To determine whether the isolated cells were able to produce cartilaginous extracellular mass, chondrogenesis was initiated using the Differentiation Basal Medium-Chondrogenic (Lonza, Basel, Switzerland), supplemented with 50 ng/mL TGF-β1 (R&D Systems, Minneapolis, MN, USA). The chondrogenic medium was exchanged twice a week for three weeks. To provide osteogenic conditions, the standard cultivation medium was substituted with the Differentiation of Basal Medium-Osteogenic (Lonza) after cells reached 100% confluence; cells were cultivated in this medium for three weeks. We also changed the medium every third day. We used histological and immunocytochemical processing to reveal signs of successful differentiation. The pellets were fixed using 10% formalin, dehydrated in ascending concentrations of ethanol, embedded in paraffin, and cut into 7 μm thick sections. After deparaffination, the chondrogenic sections were stained with blue trichrome staining modified according to Masson or processed for anti-type II collagen immunocytochemistry. We used a primary mouse IgM antibody (1:500, Sigma-Aldrich) and a Cy3TM-conjugated goat anti-mouse secondary IgM antibody. Cell nuclei were counterstained with 4′-6-diamidino-2-phenylindole (DAPI, Sigma-Aldrich). The calcium deposits were visualized in sections using von Kossa histological staining. Osteogenic sections were also processed for anti-osteocalcin immunocytochemistry. After deparaffination, samples were exposed to a primary mouse IgG antibody (1:50, Millipore, Burlington, MA, USA) and donkey anti-mouse secondary IgG antibody (1:250, Jackson ImmunoResearch Labs, West Grove, PA, USA).

Differentiation in adipocytes was induced with hMSC Adipogenic Induction medium (Lonza) and maintained with hMSC Adipogenic Maintenance SingleQuots (Lonza). Media were used subsequently and switched every three days for three weeks. For the fourth week, DPSCs were cultivated only in the hMSC Adipogenic Maintenance medium. Cultures were then fixed with 10% formalin and rinsed with 50% ethanol. Oil Red-O staining solution was applied afterwards for one hour at room temperature. The cells were observed using both a phase contrast and an inverted light microscope.

### 4.5. Statistical Analysis

All statistical analyses were performed using the statistical software GraphPad Prism 9 (San Diego, CA, USA). After normality had been ascertained using a Shapiro-Wilk test or Kolmogorov–Smirnov test, paired lineages were compared using paired *t*-test for continuous variables or a Wilcoxon matched-pairs test on ranks for nonparametric variables. Differences were considered statistically significant at *p*-values of ≤0.05.

## 5. Conclusions

We rejected the hypothesis that there is significant intra-individual variability of the DPSC profile within lineages isolated from different teeth of the same donor. We did not observe any significant effects on proliferation rate, viability, phenotype, or relive telomere length change. The only variances were seen in DPSC lineages isolated from the same donor, but the teeth were in different stages of root development. However, even in these cases, we did not find a statistical significance. The minor statistically non-significant heterogeneity in phenotype was probably because DPSC cell lines comprise heterogeneous groups of undifferentiated cells independent of the donor. We also did not observe any changes in the ability of cells to differentiate into mature cell lines—chondrocytes, osteocytes, and adipocytes.

## Figures and Tables

**Figure 1 ijms-22-13515-f001:**
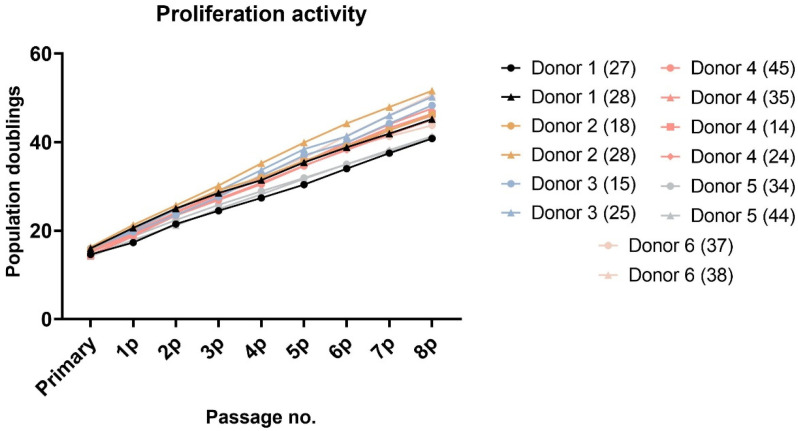
Cumulative population doublings of all paired DPSCs reached from the primary to the 8th passage. We used the formula PD = log_2_ (N_x_/N_1_) for calculating the population doublings reached in each passage. N_x_ is the total passage cell count calculated using the Z2-Counter (Beckman Coulter, Miami, FL, USA), and N_1_ is the initial cell count seeded into the culture dish (5000 cells/cm^2^). The statistical significance was calculated using a paired *t*-test; no comparison was statistically significant.

**Figure 2 ijms-22-13515-f002:**
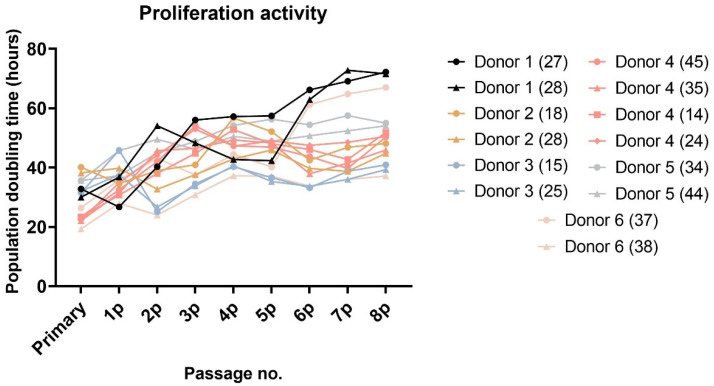
The population doubling time (PDT) in hours of all paired DPSCs reached from the primary to the 8th passage. We used the formula PDT = t/n, where t is the number of hours of cultivation per passage and n is the number of PDs in that passage. The statistical significance was calculated using a paired *t*-test; no comparison was statistically significant.

**Figure 3 ijms-22-13515-f003:**
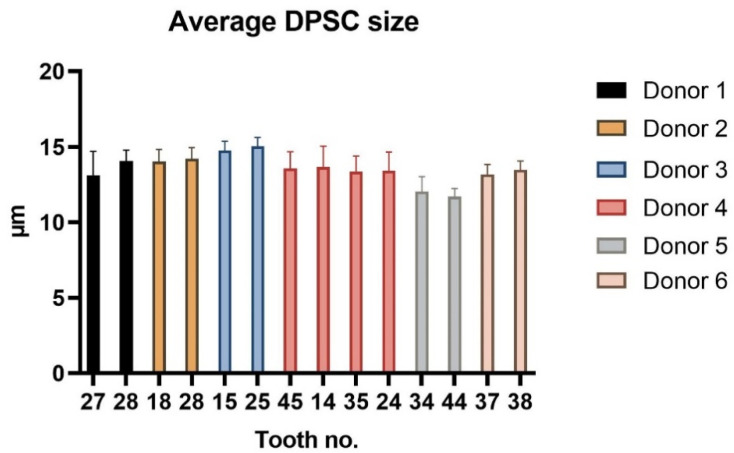
The average cell sizes of all paired DPSC lineages in μm during entire cultivation. Data are presented as a mean, and SD plotted as error bars. The statistical significance was calculated using a paired *t*-test; no comparison was statistically significant.

**Figure 4 ijms-22-13515-f004:**
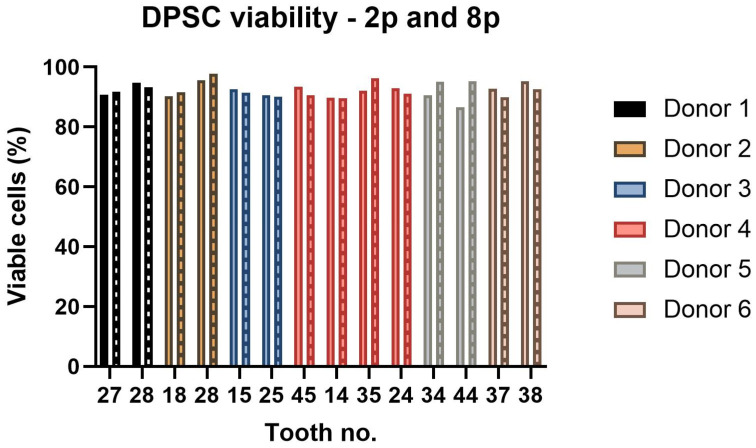
The DPSC viability of all paired lineages evaluated in the 2nd (2p) and 8th (8p) passages (boxes filled with pattern). The graph presents the percentage of viable cells. The statistical analysis was performed using a paired *t*-test; no comparison was statistically significant.

**Figure 5 ijms-22-13515-f005:**
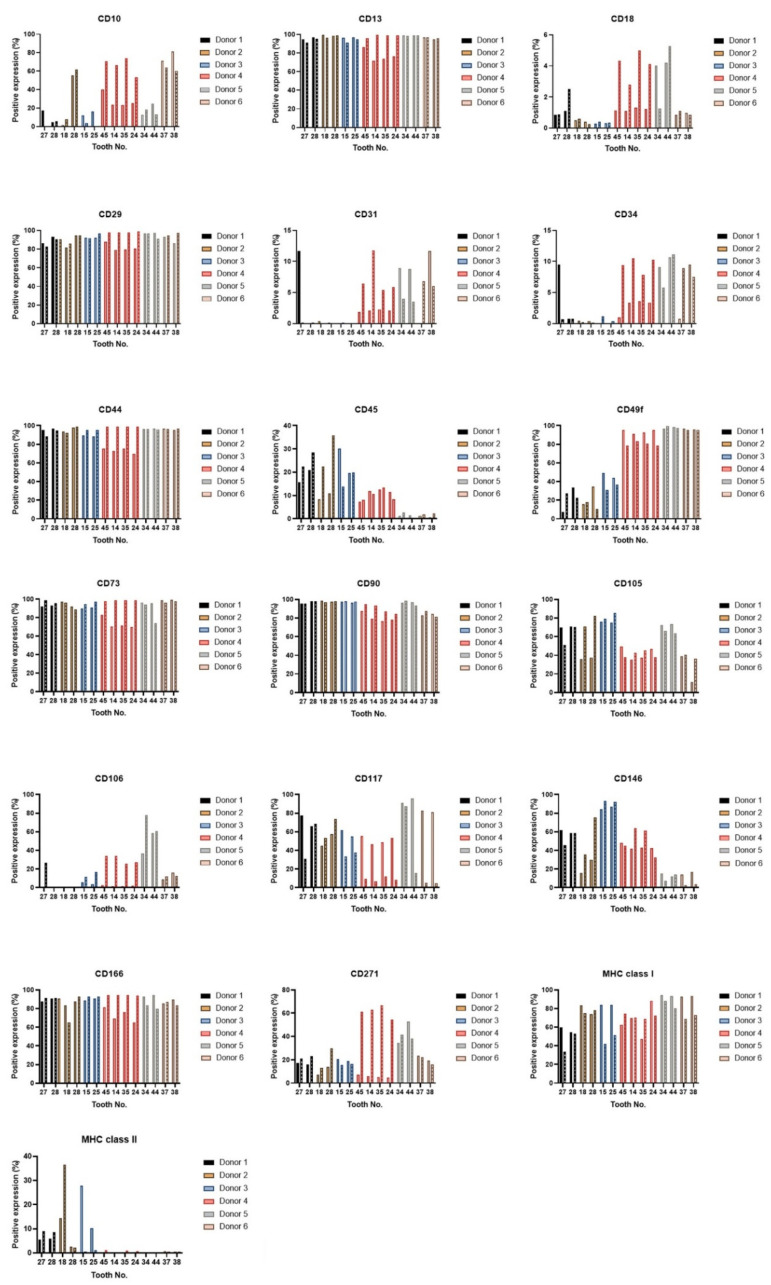
Immunophenotype profile of all paired lineages in the 3rd and 7th passages. Data for the 7th passage are depicted as boxes filled with patterns. The layouts of graphs illustrate percentages of positive cells, determined as the percentage with a fluorescence intensity greater than 99.5% of the negative isotype immunoglobulin control. After normality had been ascertained using a Shapiro–Wilk test or Kolmogorov–Smirnov test, paired lineages were compared using paired *t*-test for continuous variables, or Wilcoxon matched-pairs test on ranks for nonparametric variables. No variances were shown as statistically significant.

**Figure 6 ijms-22-13515-f006:**
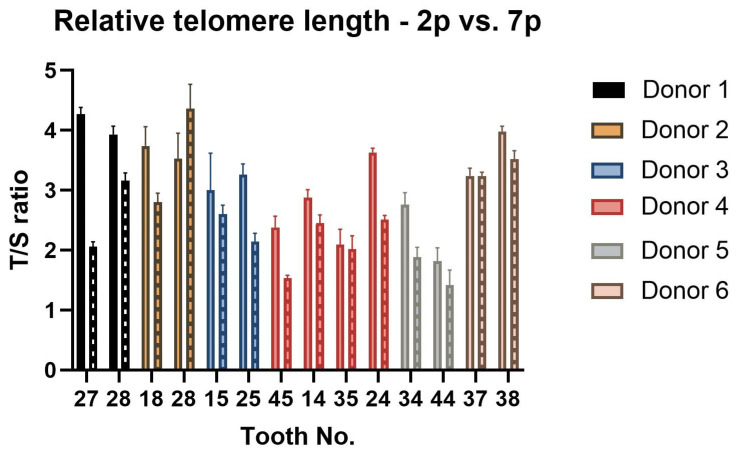
The relative telomere length change measurement between 2nd and 7th passage—T/S ratio. Data are presented as a mean, and SD plotted as error bars. Data for the 7th passage are depicted as boxes filled with patterns. The error bars were calculated for three technical replicates without biological replicates. The statistical significance was calculated using a paired *t*-test; no comparison was statistically significant. The trend of relative telomere length shortening in the 7th passage measurement was statistically significant compared to the measurement performed in the 2nd passage (*p* < 0.05).

**Figure 7 ijms-22-13515-f007:**
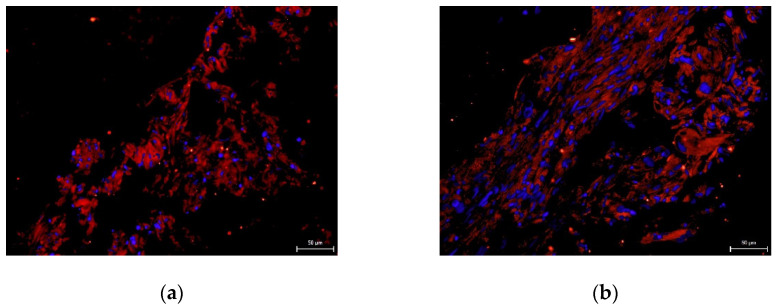
Immunocytochemical visualization of type II collagen in an extracellular matrix of paired DPSC lineages after chondrogenesis. Type II collagen shows up as fluorescent red, and stem cell nuclei fluorescent blue. Scale bar 50 μm. (**a**) Lineage 5-A; (**b**) Lineage 5-B.

**Figure 8 ijms-22-13515-f008:**
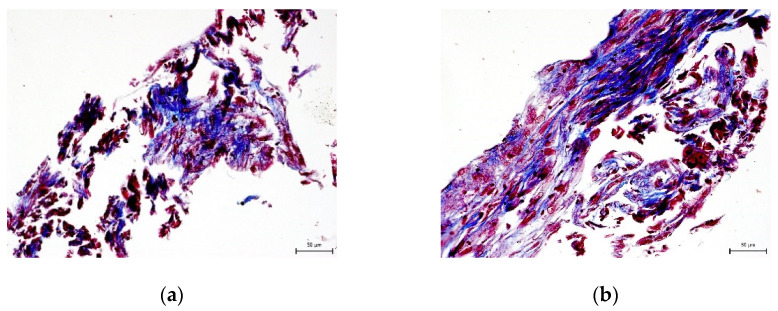
Histological visualization of collagen and procollagen in an extracellular matrix of paired DPSC lineages after chondrogenesis. After histological staining using blue Masson’s trichrome, the collagen and procollagen appear as blue areas. Scale bar 50 μm. (**a**) Lineage 5-A; (**b**) Lineaage 5-B.

**Figure 9 ijms-22-13515-f009:**
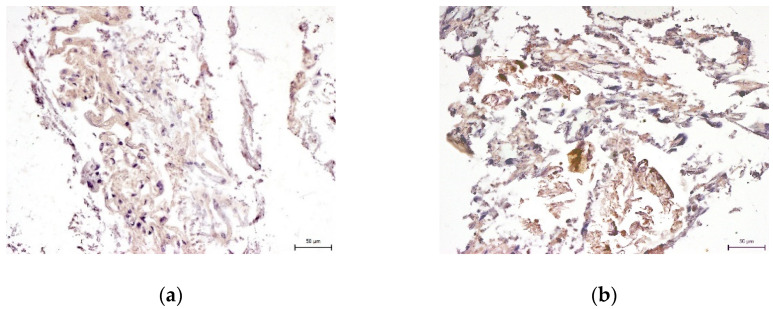
Immunocytochemical visualization of osteocalcin in the extracellular matrix of paired DPSC lineages after osteogenesis. Osteocalcin is revealed as brown areas. Scale bar 50 μm. (**a**) Lineage 5-A; (**b**) Lineage 5-B.

**Figure 10 ijms-22-13515-f010:**
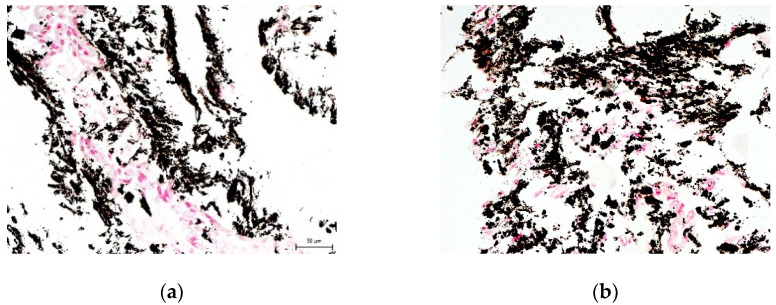
The histological detection of calcium phosphate deposits in the extracellular matrix of paired DPSC lineages after osteogenesis. Deposits are observed as black/brown spots. Scale bar 50 μm. (**a**) Lineage 5-A; (**b**) Lineage 5-B.

**Figure 11 ijms-22-13515-f011:**
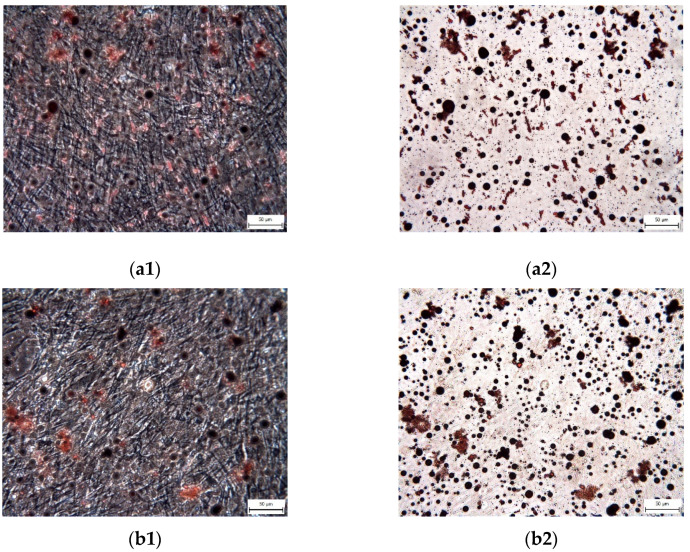
The visualization of adipose vacuoles and droplets in the extracellular matrix of paired DPSC lineages after adipogenic differentiation. After Oil Red staining, the adipose vacuoles are revealed as red areas. Scale bar 50 μm. (**a1**) Lineage 5-A (phase contrast optical microscope); (**a2**) Lineage 5-A (inverted optical microscope); (**b1**) Lineage 5-B (phase contrast optical microscope); (**b2**) Lineage 5-B (inverted optical microscope).

**Table 1 ijms-22-13515-t001:** Overview of donors’, sex, extracted teeth, and their root development, and levels of eruption.

Donor	Lineage	Sex	Age	Toth (FDI Notation)	Root Development	Eruption
1	1-A	Female	16	27	More than 1/2	Impacted
	1-B			28	Up than 1/2	Impacted
2	2-A	Female	15	18	Up to 1/2	Impacted
	2-B			28	Up to 1/2	Impacted
3	3-A	Male	13	15	More than 1/2	Impacted
	3-B			25	More than 1/2	Impacted
4	4-A	Male	15	45	Mature tooth	Erupted
	4-B			14	Mature tooth	Erupted
	4-C			35	Mature tooth	Erupted
	4-D			24	Mature tooth	Erupted
5	5-A	Female	12	34	More than 1/2	Erupted
	5-B			44	More than 1/2	Erupted
6	6-A	Male	13	37	More than 1/2	Impacted
	6-B			38	Up to 1/2	Impacted

Paired DPSC lineage isolated from different teeth of the same donor are distinguished with capital letters A, B, C, or D.

## Data Availability

Data are contained within the article.

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
