# Peer review of "Intra-Individual Variability of Human Dental Pulp Stem Cell Features Isolated from the Same Donor"

_ijms, 2021, doi:10.3390/ijms222413515_

Round 1
Reviewer 1 Report
I congrat with the Authors for the article.
This is basically a well done study.
I only have some points to ask the Authors to check.
Abstract:
- Line 13: add DPSCs in parenthesis after dental pulp stem cells since is the first time it appears in the abstract.
Main text:
- Line 77: Add a description of table 1.
References:
The following article match the topic of the study. I encourage the Authors to read it and cite it in the text in order to help the article improvement.
Genova, T.; Cavagnetto, D.; Tasinato, F.; Petrillo, S.; Ruffinatti, F.A.; Mela, L.; Carossa, M.; Munaron, L.; Roato, I.; Mussano, F. Isolation and Characterization of Buccal Fat Pad and Dental Pulp MSCs from the Same Donor. Biomedicines 2021, 9, 265. https://doi.org/10.3390/biomedicines9030265
Reviewer 2 Report
Dear Authors
I found your paper interesting and I think it can be accepted with small changes.
- the introduction does not take into account some aspects of DPSC cells such as for example, angiogenesis and the role of media conditionals. please review the introduction. Furthermore, the authors should introduce the concept of heterogeneity of the DPSC also in the introduction
- review the bibliography according to the instructions for the authors. Also, update the references with the most recent ones. references number 37 and 39 are the same. delete Number 39
- did the authors assess any gender differences? it might be interesting to review the results based on gender differences. this could bring new discoveries to light and enrich the paper.
- in table number 1 the title is missing.
- line 40. Delete the point before reference 11.
- line 270. It would be better to change the "rpm" to "x g"
- References number 37 and 39 are the same.
Reviewer 3 Report
Overall it is a good and well-written article.
Abstract: authors must indicate what the abbreviation DPSCs refers to before using it.
Lines 18-19: the authors say they found no variability in phenotype but then in lines 21 and 22 they say the heterogeneity in phenotype was due to different subpopulations. This is a contradiction. They must clarify this.
Introduction: when the authors approach pulp, in addition to the DPSCs they should refer to the different types of cells that constitute them. They should also mention the role of DPSCs in the pulp.
Line 87: the authors talk about variance, but what exactly do they mean? that the values are slightly different? In the legend of Figure 1, they state that no comparison is statistically significant, so talking about variance can be misleading.
Figure 3: how many cells were analyzed for each condition.
Figure 3: the authors refer that it is the size of the cells throughout the culture. what does this mean? Which is an average of the cell sizes in P1, P2.. through P8. Have variations in size been detected along the passages? Presentation in this way makes it impossible to perceive it.
Figure 4: Was only one sample analyzed for each group and passage? Don't the authors think that this could mislead the measurement?
Line 118: This reference and information are wrong. Please see and cite the article by Dominici, 2006. Please correct the information and reference.
Figure 5: The figure is unreadable. It has to be exchanged for a picture with larger size that allows it to be read. Authors may choose to place some graphics in supplementary material, but with a larger size.
Figure 5: Was only one sample analyzed for each group and passage? Don't the authors think that this could mislead the measurement?
Figure 5: other analyzes were performed in the 2nd and 8th or 2nd and 7th passes. This analysis was only made to the third pass. This could be a bias, as changes in markers may occur throughout the culture. Don't the authors think this is a limitation?
Section 2.4: the authors could have chosen to do extraction and quantification of Oil Red O, for example, to obtain quantitative results. Why has this not been done?
Lines 181-182: I do not agree with this information. The study was carried out to see whether there was intra-individual variation or not. If there was, it was not necessarily linked to methodological errors, but for example it could be linked to other factors such as the developmental state of the tooth.
Discussion: all patients are very young, from 12 to 16 years old. Do the authors think that if they included older patients the results would be different? They must do this discussion.
Discussion: authors should add perspectives/future studies
Lines 382-384: they cannot do this conclusion. Since they do not find phenotypic heterogeneity they cannot say that they reject this hypothesis. They don't know what would happen if they had found it. This has to be removed.
Round 2
Reviewer 3 Report
The authors did most of the proposed changes. I think that it can be published.